# CoT-Self-Instruct:
# Building high-quality synthetic data for reasoning and non-reasoning tasks

## Abstract

We propose **CoT-Self-Instruct**, a synthetic data generation method that instructs LLMs to first reason and plan via Chain-of-Thought (CoT) based on given seed tasks, and then generate a new synthetic example of similar quality and complexity. This is followed by a filtering step to select high-quality data using automatic metrics, which are then used for LLM training. In verifiable reasoning, our synthetic data significantly outperforms existing training datasets, such as s1k and OpenMathReasoning, when evaluated on MATH500, AMC23, AIME24, and GPQA-Diamond. For non-verifiable instruction-following tasks, our method surpasses the performance of both human and standard Self-Instruct training data on the AlpacaEval 2.0 and Arena-Hard benchmarks.

## 1 Introduction

The transformative rise of Large Language Models (LLMs) has initiated a substantial paradigm shift in the domain of deep learning (Zhang et al., 2023; Guo et al., 2023; Long et al., 2024). The development of such models emphasizes scale, and relies heavily on large volumes of high-quality data (Gandhi et al., 2024; Abdin et al., 2024). However, acquiring such data from human sources can often be challenging or even impractical due to factors such as high costs, data scarcity, and privacy concerns (Kurakin et al., 2023). Furthermore, several studies (Hosking et al., 2023; Singh et al., 2023; Gilardi et al., 2023) have pointed out that human-generated data, being inherently prone to biases and errors, may not always be ideal for model training or evaluation. In this context, synthetic data emerges as a viable alternative for obtaining high-quality datasets.

Synthetic data is artificially generated to replicate the characteristics and patterns of real-world data. One innovative approach to creating such data is the Self-Instruct method (Wang et al., 2022a), which utilizes LLMs themselves to generate instruction-following examples. This method begins by selecting a small set of seed instruction-following samples, which are then used to prompt LLMs to produce additional demonstrations in a similar format. Since then, a number of variants have been introduced that increase the complexity of queries (Liu et al., 2023; Zeng et al., 2024), maintain semantic diversity (Ding et al., 2023), scale the synthetic data (Yuan et al., 2023), and use these methods in self-improvement loops (Yuan et al., 2024). However, a significant challenge with these approaches is to ensure the quality and effectiveness of the generated data for language model training. Overall, generating high-quality synthetic data and optimizing its use for both reasoning and non-reasoning tasks still remains insufficiently understood.

In this paper, we present Chain-of-Thought(CoT)-Self-Instruct, a method that both (i) uses reasoning to help create high-quality synthetic data; and (ii) self-filters the created data to only keep the highest quality ones; see Figure 1. Reasoning with CoT allows the model to analyze the given few-shot examples and plan the *generation of challenging examples*, while ensuring their logical validity. We show the efficacy of this approach for creating both verifiable reasoning data and non-verifiable instruction following tasks, where in both cases using CoT outperforms those generated without CoT. To curate high-quality verifiable data, we introduce Answer-Consistency, whereby we first create both a synthetic instruction and a target answer. Then we discard examples where this target answer does not match the majority vote solution of the LLM, with the assumption that those examples are either incorrectly labeled or too difficult. For non-verifiable data, we use the recent

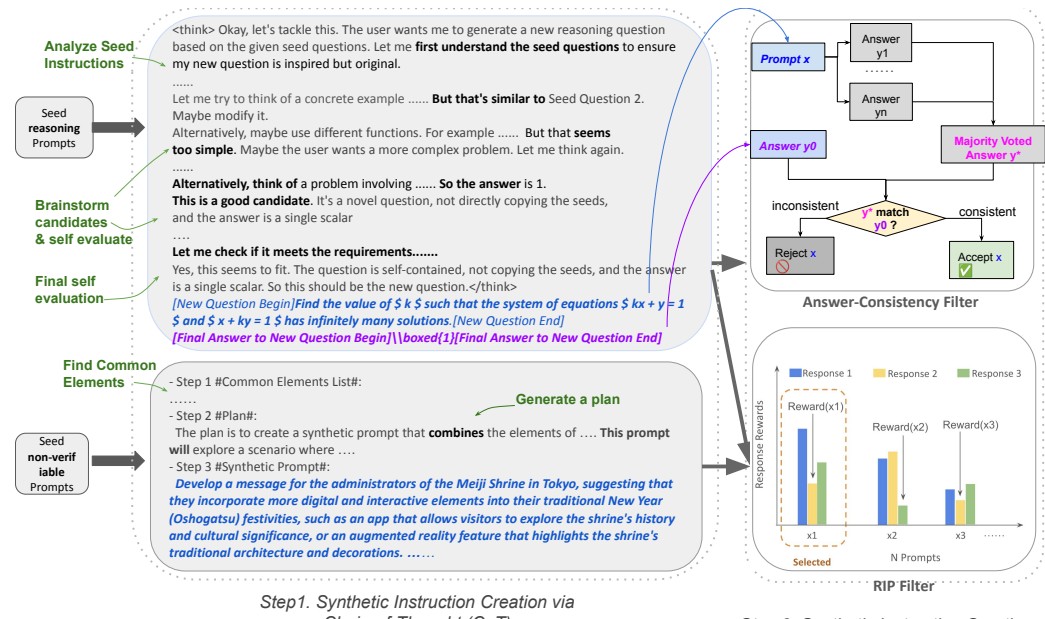

Figure 1: **Overview of CoT-Self-Instruct.** Our method first prompts LLMs to reason and generate new instructions given seed instructions, followed by automatic curation of high-quality data using either Answer-Consistency for verifiable reasoning tasks, or RIP (Yu et al., 2025) for non-verifiable tasks.

Rejecting Instruction Preferences (RIP) (Yu et al., 2025) method, which measures the quality of instructions based on the distribution of reward model scores assigned to sampled LLM responses. In both cases, filtering provides further gains. For reasoning tasks, CoT-Self-Instruct generated training data outperforms Self-Instruct and existing curated datasets such as s1k (Muennighoff et al., 2025) and OpenMathReasoning (Moshkov et al., 2025) when the trained LLM is evaluated on MATH500, AMC23, AIME24 and GPQA-Diamond. For non-verifiable tasks, it outperforms human data from WildChat (Zhao et al., 2024) and Self-Instruct synthetic data, whether filtered or not, on both the AlpacaEval 2 and ArenaHard benchmarks. Overall, models trained with CoT-Self-Instruct generated training data provided the best results among the methods we tested.

## 2 RELATED WORK

**Synthetic data generation**  Synthetic data is produced using algorithms (Saxton et al., 2019), generative models (Borisov et al., 2022; Meng et al., 2022), or simulations (Vezhnevets et al., 2023), rather than being directly created by humans (Liu et al., 2024). It presents a promising solution for training models, particularly in scenarios where real-world data is scarce, expensive, or difficult to obtain. Self-Instruct (Wang et al., 2022a) first proposed a framework that prompts a language model with seed data as few-shot examples in order to generate new synthetic data. Such data has subsequently been used to self-train language models, e.g. in the Self-Rewarding framework (Yuan et al., 2024; Wu et al., 2024). Evol Instruct (Zeng et al., 2024) proposed to increase instruction complexity by letting the language model rewrite the original instructions with added complexity. Other specific methods of creating complex synthetic data have been proposed, such as multi-hop question answering (Lupidi et al., 2024) or difficult reasoning questions (Guo et al., 2025b; Yuan et al., 2025), both grounded on real documents. Synthetic data has also been used to help train agents (Zhao et al., 2025; Zhou et al., 2025) and tool-use models (Mekala et al., 2024), as well as for rewriting pre-training data (Maini et al., 2024; Nguyen et al., 2025).

**Synthetic data selection**  Data selection is a critical component for post-training with synthetic data (and for data in general). Previously, LLM training was regarded as largely dependent on the

size of available training data (Mishra et al., 2021; Wei et al., 2021; Wang et al., 2022b). More recent work, however, has revealed that training on a smaller yet higher-quality curated set of instructions tends to be more effective in improving models' both instruction following and reasoning capabilities (Zhou et al., 2024; Chen et al., 2024; Muennighoff et al., 2025; Ye et al., 2025). In addition to preprocessing techniques such as deduplication of similar instructions using similarity metrics such as ROUGE-L similarity score (Wang et al., 2022a) or clustering (Chen et al., 2023), as language models become more powerful, data curation can also be facilitated by using LLMs themselves as a quality judge. Recent work studies employing powerful language models to measure the complexity, diversity and quality of instructions (Lu et al., 2023; Chen et al., 2024; Touvron et al., 2023; Dubey et al., 2024; Li et al., 2023a). The success of RLHF for post-training (Stiennon et al., 2020; Rafailov et al., 2024) has attracted more attention to collecting large-scale and high-quality preference data. Most work involving preference optimization employs existing methods derived from pretraining and instruction-tuning (Touvron et al., 2023; Muennighoff et al., 2025), such as deduplication, clustering, quality classifiers or filtering heuristics. Rejecting Instruction Preferences (RIP) (Yu et al., 2025) is a recent effective method that leverages reward model scores assigned to LLM generated responses when filtering out low-quality instructions. For verifiable reasoning tasks, Self-Consistency filtering (Prasad et al., 2024) has also been shown to be a high-quality curation method by rejecting instructions where LLM solutions show low agreement, which suggests the task is either incorrectly labeled or too difficult.

## 3  CHAIN-OF-THOUGHT(CoT)-SELF-INSTRUCT

CoT-Self-Instruct is an approach to generate high-quality synthetic data for training using reasoning. We first assume access to a language model, and a small amount of high-quality human-annotated seed data. We consider both verifiable reasoning domains, and non-verifiable general instruction following. Our approach involves two stages:

1. Synthetic Instruction Creation with Chain-of-Thought (CoT): given sample human-annotated seed instructions, we instruct the LLM to reason step-by-step to come up with instructions of similar complexity and domain.

2. Synthetic Instruction Curation: we curate the generated synthetic data to keep only high-quality instructions for self-training.

We then train LLMs using the generated high-quality synthetic instructions. Below, we describe each stage in turn.

### 3.1  SYNTHETIC INSTRUCTION CREATION VIA CoT

The process of CoT-Self-Instruct data creation starts with a small set of seed instructions as the instruction pool. Multiple instructions are sampled at random from the instruction pool, and then used to few-shot prompt a language model to generate a series of intermediate reasoning steps, followed by a new instruction. Unlike standard Self-Instruct (Wang et al., 2022a), which prompts the model to directly write new instructions given a list of seed instructions, we first ask the model to carefully analyze the given instructions, such as their domain, complexity and purpose, and to reflect on what makes them high-quality instructions. After this analysis, the LLM then reasons step-by-step to come up with a plan to generate a new self-contained instruction that is of similar quality and complexity as the given seed instructions, and ultimately outputs the final synthetic instruction satisfying these requirements in a strict answer format.

**Verifiable reasoning tasks**  For reasoning tasks where there is a deterministic answer which we can compare against to generate verifiable rewards during training, we instruct the LLM to use reasoning to generate both an instruction and its verifiable answer at the same time. This allows the model to simultaneously solve the problem while creating it step-by-step, which can be easier than solving the final problem directly. The prompt we used for CoT-Self-Instruct on reasoning tasks is given in Figure 2, which has a strict formatting rule to allow easy separation of an instruction from its target answer.

Figure 2: CoT-Self-Instruct instruction generation template for verifiable reasoning tasks. In this template, the LLM is prompted to generate both the question and its corresponding answer—complete with chain-of-thought reasoning—in a single pass.

---

You are a **reasoning question generator assistant**. Your goal is to create a novel, and challenging reasoning question. You are provided the following seed questions:

Seed Question 1: **{INSTRUCTION 1}**
Seed Question 2: **{INSTRUCTION 2}**

Your task is to:
1. Write a brand-new, self-contained reasoning question that meets the following requirements:
(a) The question draws inspiration from the seed question without copying it verbatim, remaining novel and of comparable difficulty.
(b) The question's final answer should be a single, unambiguous scalar value (e.g., an integer, reduced fraction, exact radical), or another answer type that can be verified in one step (e.g., 'yes/no,' a choice from A to D).
2. Then reason step by step, solve the new question and format your output as follows:
[New Question Begin]{your_generated_question}[New Question End]
[Final Answer to New Question Begin]\boxed{your_final_answer}[Final Answer to New Question End]

---

**General instruction following tasks**   For tasks involving general instruction-following with open-ended responses, we direct the LLM to use reasoning to generate an instruction only, and not to include a specific response. In these instances, later during training on this synthetic data, we utilize a reward model to assess policy model responses, eliminating the need for a reference answer. The prompt we used for CoT-Self-Instruct on general instruction following tasks is given in Figure 3. Seed instruction pools for instruction-following typically include various different domains. When selecting few-shot samples, for example, combining instructions from storytelling and coding could result in unnatural synthetic instructions. To address this, we propose to first group seed instructions into categories by their domain. For each synthetically generated instruction, we first sample a category, and then select instructions from within that category as few-shot examples.

## 3.2 SYNTHETIC INSTRUCTION CURATION

Even with the strongest language models, not all generated synthetic instructions are well-defined and answerable, or are effective in base model self-training. We therefore apply a curation step to select higher-quality synthetic instructions from the pool of generated data for final post-training with RL.

**Verifiable reasoning tasks**   We propose *Answer-Consistency* to filter and retain only high-quality data. Given the task instruction, we first instruct the LLM to generate $K$ responses and take the majority vote over the final answers. We then reject the data example and remove it from the training pool if the majority vote does not match the target answer included in the synthetic data example generated by CoT-Self-Instruct (i.e., via Figure 2). Because the target answer from CoT-Self-Instruct is generated with extensive reasoning steps during instruction writing and question-answering (Figure 2), it has access to more information (e.g. the step-by-step creation process of a problem) compared to an LLM answer generated at inference time given only the problem. Hence, comparing if the answers match gives an extra layer of filtering. We confirm in our experiments that Answer-Consistency is superior to standard Self-Consistency filtering (Prasad et al., 2024).

Figure 3: CoT-Self-Instruct instruction generation template for general instruction following tasks.

> You are a **prompt generator assistant**. Your goal is to create diverse and creative synthetic prompts.
>
> Please follow the steps below to create synthetic prompts.
>
> Step 1: Carefully read #Prompt 1# and #Prompt 2#. Identify and list all the common elements between these two prompts. If no common elements are found, list the main elements from each prompt.
> Step 2: Develop a comprehensive plan based on the #Common Elements List# or #Main Elements List# from Step 1. This plan will guide the generation of new synthetic prompts that are similar to the original prompts.
> Step 3: Execute the plan step by step and provide one #Synthetic Prompt#.
>
> Please reply strictly in the following format:
> - Step 1 #Common Elements List# or #Main Elements List#:
> - Step 2 #Plan#:
> - Step 3 #Synthetic Prompt#:
>
> #Prompt 1#: {**INSTRUCTION 1**}
> #Prompt 2#: {**INSTRUCTION 2**}

**General instruction following tasks** For non-verifiable tasks, the Answer-Consistency method is not applicable as we do not generate target responses when creating synthetic examples for open-ended questions. Instead, we employ ideas from the Rejecting Instruction Preferences (**RIP**) method as proposed by Yu et al. (2025). In this method, for a given task instruction, $K$ responses are generated, and each response is evaluated using a reward model (RM), resulting in a rating for each response. The filtering process is then based on the distribution of these ratings. We use the lowest rating among these $K$ responses to represent the overall score of a given synthetic instruction. While Yu et al. (2025) set a global threshold for filtering over these scores, we notice a topic distribution shift if we perform such global filtering, as each topic category has different score distributions. Therefore, we slightly modify RIP by sampling multiple instructions from each few-shot prompt, then selecting the one with the highest score. Notably, the RIP approach can also be applied to verifiable tasks, and we conduct experiments in that context as well.

## 3.3 SELF-TRAINING WITH SYNTHETIC DATA

After generating the synthetic training data, we can then use them to conduct RL training in order to improve the downstream performance of an LLM. We compare the performance of such self-trained LLMs with models trained on human-annotated data and on curated seed instructions in reasoning and general instruction following domains respectively. For verifiable reasoning tasks, we train with GRPO (Shao et al., 2024), and for general instruction following we consider both offline DPO (Rafailov et al., 2024) and online DPO, which can perform much better, see e.g. Lanchantin et al. (2025).

## 4 EXPERIMENTAL SETUP

We study the effectiveness of our synthetic data generation approach for reasoning and general instruction following domains along the following two axes: synthetic instruction generation, and instruction curation.

## 4.1 VERIFIABLE REASONING TASKS

**Seed instructions**    We use the s1k reasoning instructions from (Muennighoff et al., 2025) as our seed tasks. The s1k dataset contains 1,000 high-quality, diverse, and challenging reasoning questions, which are on par with those found in the original 59,000 sample dataset according to their paper. To conduct self-training with verifiable rewards, we select a subset of s1k consisting of 893 verifiable reasoning instructions by filtering out theorem-proving questions and only keeping those that yield a scalar, single-valued, or simple closed-form answers that can be easily verified (such as $1$, $A$, $False$, $\frac{2(n-1)(n-2)}{n(n+1)}$). We then use this subset as the seed instruction pool to generate more verifiable reasoning instructions.

**Instruction generation**    The CoT-Self-Instruct template is given in Figure 2. To evaluate how CoT-Self-Instruct compares to baselines for generating verifiable reasoning tasks, we apply these methods to the following models: Qwen3-4B-Base, Qwen3-4B with Think mode and Qwen3-4B with NoThink mode (Yang et al., 2025). We generate up to 10,000 instructions using temperature=0.7 and top-p=0.8 for Qwen3-4B-Base and Qwen3-4B (NoThink mode), and temperature=0.6 and top-p=0.95 for Qwen3-4B (Think mode).

**RLVR training**    All our reasoning experiments use GRPO training initialized from Qwen3-4B-Base with reinforcement learning from rule-based verifiable rewards (RLVR). For hyperparameters, we use a cosine learning rate scheduler with a peak value of $1e-6$ and adopt the AdamW optimizer for the policy model. We set the number of training epochs to 40 with a batch size of 128. For rollouts, we sample 16 rollouts for each prompt with temperature=0.6 and top-p=0.95, with a maximum length of 4096 tokens. All GRPO experiments are conducted with VeRL(Sheng et al., 2024) and Math-Verify [1] as a verifier.

**Baselines & variations**    We compare our method CoT-Self-Instruct with standard Self-Instruct with no CoT for generating either instruction or target answer (template given in Appendix Figure 7). We also compare to using no CoT for generating the instruction, but using CoT to generate the target (template given in Appendix Figure 6). These variations test the importance of using CoT for both instruction and target when generating synthetic training examples. As further baselines, we also train on the original s1k instructions rather than synthetic data, in that case we use the publicly available DeepSeek R1 (Guo et al., 2025a) thinking solution from simplescaling/s1K-1.1 to build targets. We also compare to training on OpenMathReasoning which consists of 10k instructions (Moshkov et al., 2025) with publicly available solutions by DeepSeek-R1 and QwQ-32B. We also explore some alternative ways of filtering data or constructing target labels, explored as variations on our main experiments:

- Self-Consistency filtering: generate $K$ responses with random seeds and then select the majority-voted answer as the target or reject the example if the majority answer receives fewer votes than a given threshold (50% in our main experiments).

- RIP filtering: we use the infly/INF-ORM-Llama3.1-70B (Minghao Yang, 2024) reward model.

- Best-of-$K$ targets: constructing targets by selecting the highest scored answer out of $K$ responses using INF-ORM-Llama3.1-70B RM.

**Evaluation**    We evaluate our trained models on math tasks using MATH500 (Hendrycks et al., 2021; Lightman et al., 2023), AIME 2024 and AMC 23. We also evaluate on GPQA Diamond (Rein et al., 2024), which consists of challenging science questions. We use temperature=0.6 and top-p=0.95 to generate predictions. For each problem, we generate $N = 16$ solutions and report the average accuracy.

---

[1] https://github.com/huggingface/Math-Verify

## 4.2 GENERAL INSTRUCTION FOLLOWING

**Seed instructions** As our seed examples, we use the Wildchat-RIP-Filtered-by-8b-Llama dataset[2], which includes 4k high-quality instructions filtered from 20k raw Wildchat samples. We prompt the LLama 3.3-70B-Instruct model to classify each seed instruction into one of the following 8 categories: Writing & Storytelling, Technical & Programming, Creative & Design, Data & Analysis, Education & Research, Communication & Support, Business & Marketing, and Miscellaneous.

**Instruction generation** We evaluate how CoT-Self-Instruct compares to the baselines for generating non-verifiable instruction-following tasks by training LLaMA 3.1-8B-Instruct on the generated data. The CoT-Self-Instruct template is given in Figure 3, and the baseline Self-Instruct method is given in Appendix Figure 4. We also experiment with an approach that lies between the two methods by generating a shorter, rather than a longer, CoT, as shown in Appendix Figure 5. We prepare a set of 5,000 few-shot prompts, each consisting of two randomly selected seed examples from the same category. For RIP filtering, we generate 32 synthetic instructions from each few-shot prompt, and keep the one with the highest RIP score, resulting in 5,000 synthetic instructions in total. We use the Athene-RM-8B[3] reward model for the RIP scoring.

**DPO training** We train via DPO starting from Llama 3.1-8B-Instruct, leveraging the `fairseq2` library (Balioglu, 2023). We use a batch size of $64$ and a learning rate of $1e-6$ with a dropout rate of 0.0 and a $\beta$ value of 0.1 throughout the experiments. For each instruction, we generate 64 responses. These responses are then annotated with Athene-RM-8B to select preference pairs. Compared to human instructions, our synthetic instructions tend to be more complex, resulting in longer average response lengths, which can lead to length explosion. This occurs because, during DPO training, the evaluation judge often favors longer responses, potentially causing response lengths to increase over time (Yuan et al., 2024). To mitigate this issue, we adopted the approach outlined by Wu et al. (2024), which involves combining the reward score with length information to determine the preferred response. This method ensures that shorter responses are selected when scores are similar. We applied a length normalization coefficient of 0.2 for the length-normalized reward. This is applied for all methods, in each case constructing 5,000 DPO pairs.

**Online DPO training** We also experiment with online DPO training by following the settings described by Lanchantin et al. (2025). We use the default sampling parameters (temperature=1.0, top-p=1.0) to generate exploration rollouts. We train models using the `fairseq2` library (Balioglu, 2023), where model inference is performed with the `vllm` library (Kwon et al., 2023).

**Evaluation** To evaluate the helpfulness and quality of responses, we employ AlpacaEval 2.0 (Li et al., 2023b; Dubois et al., 2024) and Arena-Hard (Li et al., 2024b;a). These are robust instruction-following benchmarks that show a strong correlation with user preferences. Originally, AlpacaEval used GPT-4 Preview (11/06) as the judge, while Arena-Hard utilized GPT-4.1 for its leaderboard. However, since we do not have access to these specific OpenAI API versions, we conduct our tests using two alternative (and newer) judges: GPT-4-turbo and GPT-4o. For generating responses, we set the decoding temperature to 0.6 and the top-p to 0.9, aligning with the commonly used values of the seed model in our study. Our validation set, used for checkpoint selection, is based on a held-out set of 470 examples, comprising 253 validation examples from Li et al. (2023a) and 218 Evol-Test set examples from Xu et al. (2023).

## 5 EXPERIMENTAL RESULTS

Our main results are given in Table 1 for reasoning tasks and Table 2 for non-reasoning tasks. Various other variations and ablations are given in the Appendix.

---

[2] https://huggingface.co/datasets/facebook/Wildchat-RIP-Filtered-by-8b-Llama
[3] https://huggingface.co/Nexusflow/Athene-RM-8B.

Table 1: **CoT-Self-Instruct results on reasoning tasks**, compared to baselines, when fine-tuning Qwen3-4B-Base with GRPO. For Self-Instruct and CoT-Self-Instruct, the synthetic data (including targets) is constructed with Qwen3-4B. We report pass@1 averaged over 16 seeds. CoT-Self-Instruct generates synthetic data that outperforms existing training sets and the Self-Instruct method, particularly when applying our data filtering methods.

| | # Train | MATH 500 | AIME 24 | AMC 23 | GPQA Diamond | Avg. ↑ |
|---|---|---|---|---|---|---|
| Qwen3-4B-Base (Zero-Shot) | - | 67.4 | 10.6 | 42.0 | 24.2 | 36.1 |
| *s1k questions* + (R1) gold label | 893 | 68.6 | 18.5 | 51.3 | 40.1 | 44.6 |
| *OpenMathReasoning questions + gold label* | 10000 | 79.0 | 13.3 | 62.5 | 35.4 | 47.5 |
| **Self-Instruct** questions + targets | 5000 | 74.5 | 9.8 | 47.7 | 39.0 | 42.7 |
| **Self-Instruct** questions + CoT-generated targets | 5000 | 81.1 | 16.3 | 58.1 | 42.5 | 49.5 |
| + Self-Consistency Filter | 3467 | 83.6 | 18.5 | 68.5 | 44.1 | 53.6 |
| + RIP Filter | 2254 | 84.5 | 21.2 | 65.9 | 45.5 | 54.5 |
| **CoT-Self-Instruct** | 5000 | 84.9 | 20.4 | 62.2 | 44.4 | 53.0 |
| + Self-Consistency Filter | 4034 | 85.2 | 22.5 | 67.8 | 44.9 | 55.1 |
| + RIP Filter | 2419 | 85.7 | 24.4 | 70.5 | 44.4 | 56.2 |
| + Answer-Consistency Filter | 2926 | 86.5 | 24.6 | 72.3 | 45.5 | 57.2 |
| + Answer-Consistency Filter (more data) | 10000 | **86.7** | **26.7** | **73.8** | **47.4** | **58.7** |

Table 2: **CoT-Self-Instruct results on general instruction following tasks**, comparing to baselines, fine-tuning LLama 3.1-8B-Instruct with offline and online DPO. CoT-Self-Instruct generates synthetic data that outperforms human-written instruction training sets and the Self-Instruct method, particularly when applying our data filtering methods. Both AlpacaEval 2 and ArenaHard are evaluated with two kinds of judge: GPT-4 Turbo and GPT-4o, with similar conclusions.

| | Training method | AlpacaEval LC Winrate | | ArenaHard Score | | Avg. ↑ |
|---|---|---|---|---|---|---|
| | | GPT-4 Turbo | GPT-4o | GPT-4 Turbo | GPT-4o | |
| LLama 3.1-8B-Instruct | DPO | 27.3 | 21.3 | 32.0 | 27.8 | 27.1 |
| **Human instructions** (WildChat) | DPO | 49.1 | 43.0 | 52.7 | 42.6 | 46.8 |
| + RIP Filter | DPO | 57.6 | 44.5 | 59.1 | 41.7 | 50.7 |
| **Self-Instruct** | DPO | 52.9 | 46.0 | 51.8 | 39.2 | 47.4 |
| + RIP Filter | DPO | 55.2 | 46.1 | 55.6 | 39.5 | 49.1 |
| **CoT-Self-Instruct** | DPO | 58.5 | 48.6 | 62.0 | 46.7 | 53.9 |
| + RIP Filter | DPO | 63.2 | 49.4 | 60.2 | 45.8 | 54.7 |
| **Human instructions** (Wildchat) | Online DPO | 80.1 | 62.7 | 64.4 | 45.5 | 63.1 |
| **CoT-Self-Instruct** + RIP | Online DPO | **83.2** | **68.7** | **67.3** | **49.3** | **67.1** |

## 5.1 REASONING TASKS

**Synthetic instructions generated by CoT-Self-Instruct outperform Self-Instruct**   In Table 1 where the Qwen3-4B-Base models are GRPO trained on Qwen3-4B generated instructions and targets, CoT-Self-Instruct achieves an average accuracy of 53.0%, outperforming Self-Instruct which yields 42.7% (both without data filtering). If we use Self-Instruct without CoT to generate instructions, but then use CoT to generate targets, we can achieve 49.5%, still inferior to CoT-Self-Instruct which uses CoT reasoning to generate both instructions and target answers. As shown in Appendix Table 8, similar trends are observed when training on Qwen3-4B-base, rather than Qwen3-4B, generated data. This highlights the importance of CoT thinking when generating reasoning instructions.

**Filtered CoT-Self-Instruct outperforms filtered Self-Instruct**   Applying filtering methods to CoT-Self-Instruct and Self-Instruct improves both methods, despite the overall amount of training data decreasing, see Table 1. This suggests that it is better to have less high-quality synthetic data than more lower-quality data. However, we find that CoT-Self-Instruct still maintains its advantage over Self-Instruct, whichever filtering method is used. E.g. with Self-Consistency Filtering, Self-Instruct with CoT targets improves from 49.5% → 53.6%, while CoT-Self-Instruct improves from 53.0% → 55.1%. Similar findings are observed with RIP filtering as well. We find our proposed Answer-Consistency filtering, where answers are produced while generating an instruction with CoT, outperforms Self-Consistency filtering, which uses majority vote alone, with an average performance improvement from 55.1% → 57.2%.

**High quality synthetic instructions generated by CoT-Self-Instruct significantly outperform seed instructions and other publicly available reasoning instructions**  CoT-Self-Instruct outperforms s1k, as shown in Table 1, where the model trained on 2926 filtered examples using CoT-Self-Instruct achieve 57.2%. This is much higher than the 44.6% achieved with s1k instructions using R1 labels (and s1k results are even lower, 43.8%, with Qwen3-4B labels, see Appendix Table 5). Filtering CoT-Self-Instruct to the same training size as s1k yields 54.2%, still significantly higher, see Appendix Table 3. Our method also outperforms 10k OpenMath-Reasoning questions with gold labels, which gives 47.5%. Increasing the CoT-Self-Instruct with Answer-Consistency filtering data to 10k improves results further with an average of 58.7%. Overall, CoT-Self-Instruct with Answer-Consistency filter gives the best performance among all existing datasets or synthetic data construction methods tested.

**Using other models or methods to generate targets yields similar conclusions**  Our main experiments use Qwen3-4B-Base models GRPO trained on Qwen3-4B generated instructions and targets. We also experiment with several other settings. These include using Qwen3-4B-Base to generate targets (Appendix Table 8), Majority-vote to generate targets (Appendix Table 6), and Best-of-$K$ using a reward model to generate targets (Appendix Table 7). While each variant gives different overall maximum perfomance, in each case we see the same trend that CoT-Self-Instruct with Answer-Consistency is superior to Self-Instruct and other competing baselines.

## 5.2 GENERAL INSTRUCTION FOLLOWING TASKS

**Synthetic instructions generated by CoT-Self-Instruct outperform Self-Instruct**  For nonreasoning tasks, allowing the model to create a plan beforehand with CoT-Self-Instruct also significantly enhances the quality of synthetic data, see Table 2. Averaged over AlpacaEval 2 and ArenaHard, CoT-Self-Instruct achieves an average of 53.9 vs. Self-Instruct's 47.4, both without filtering and trained with DPO. We also observe that asking for longer CoT reasoning chains provides more gains than shorter CoTs (see Appendix Table 11), further emphasizing the need for reasoning even when producing synthetic data for non-verifiable general instruction following tasks.

**RIP filtering improves CoT-Self-Instruct results further**  Applying the RIP filter to each method is proven to be effective across all types of synthetic generation methods tested. It boosts the CoT-Self-Instruct results from 53.9 → 54.7. RIP also improves Self-Instruct as well, from 47.4 → 49.1, but still underperforming CoT-Self-Instruct. We can also apply RIP filtering to human instructions from WildChat in a similar manner. In this case we actually see a larger boost, from 46.8 → 50.7. We attribute this to human data being relatively noisy compared to synthetic data, which can make filtering more important.

**High quality synthetic instructions generated by CoT-Instruct significantly outperform human instructions**  Our best performing DPO-trained model is achieved by using CoT-Self-Instruct with RIP data filtering, yielding 54.7. This outperforms Llama 3.1-8B-Instruct (27.1) or training on human instructions from WildChat with (46.8) or without RIP data filtering (50.7). We also performed experiments with online DPO, which improved results further. In that setting, human instructions from WildChat obtain 63.1 while CoT-Self-Instruct+RIP obtains 67.1. Overall, we find CoT-Self-Instruct with RIP filtering to yield the best performance over all existing datasets or synthetic data construction methods tested.

## 6 CONCLUSION

In this paper, we propose CoT-Self-Instruct, a synthetic data creation and curation pipeline that instructs LLMs to plan and reason to come up with new synthetic instructions given seed examples, and then filters them for quality, either using Answer-Consistency for verifiable tasks or RIP filtering when they are not verifiable. We show that applying our method improves models' abilities in both reasoning and non-reasoning domains by creating high quality synthetic instructions for RL training, surpassing existing seed human-annotated instructions and public training sets on challenging benchmarks.

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

Figure 4: Self-Instruct instruction generation template for general instruction following tasks.

> Below are sample tasks from user.
> 1. <begin>**{INSTRUCTION 1}**</end>
> 2. <begin>**{INSTRUCTION 2}**</end>
>
> Come up with one new task, wrapped with <begin>and </end>

Figure 5: Short CoT instruction generation template for general instruction following tasks.

> Below are sample tasks from user.
> 1. <begin>**{INSTRUCTION 1}**</end>
> 2. <begin>**{INSTRUCTION 2}**</end>
>
> Come up with one new task, wrapped with <begin>and </end>. Please provide your Chain-of-Thought first and then provide the new generated task.

# 7 APPENDIX

We report results when matching the training size to 893 the same as our seed tasks in Table 3.

Table 3: **CoT-Self-Instruct results on reasoning tasks with same size training sets**, comparing to baselines, fine-tuning Qwen3-4B-Base with GRPO. For Self-Instruct and CoT-Self-Instruct the synthetic data (including targets) is constructed with Qwen3-4B. We report pass@1 averaged over 16 seeds.

| | # Train | MATH 500 | AIME 24 | AMC 23 | GPQA Diamond | Avg. ↑ |
|---|---|---|---|---|---|---|
| Qwen3-4B-Base (Zero-Shot) | - | 67.4 | 10.6 | 42.0 | 24.2 | 36.1 |
| *s1k Instructions* + (R1) Gold Label | 893 | 68.6 | 18.5 | 51.3 | 40.1 | 44.6 |
| **Self-Instruct** | 893 | 80.5 | 17.2 | 57.3 | 41.3 | 49.1 |
| + Self-Consistency Filter | 893 | 81.9 | 20.0 | 62.8 | 41.5 | 51.5 |
| + RIP Filter | 893 | 82.7 | 21.5 | 61.4 | 43.1 | 52.2 |
| **CoT-Self-Instruct** | 893 | 82.4 | 19.8 | 60.0 | 41.3 | 50.9 |
| + Self-Consistency Filter | 893 | 83.2 | 22.7 | 65.1 | 41.6 | 53.1 |
| + RIP Filter | 893 | 83.0 | 21.0 | 63.9 | 42.9 | 52.7 |
| + Answer-Consistency Filter | 893 | **83.7** | **23.1** | **66.1** | **44.1** | **54.2** |

We further compare CoT-Self-Instruct with other templates on reasoning tasks:

- Self-Instruct-Then-Solve (NoCoT): prompting LLMs to first generate a question then an answer to its own generated question, without any thinking or CoT, see Figure 9.
- CoT-Self-Instruct (NoSolve): prompting LLMs to reason step-by-step to generate a question, without giving the "reference" answer, see Figure 8.

We report additional results with varying prompt templates below.

Figure 6: Self-Instruct (standard, without CoT) instruction generation template for verifiable reasoning tasks. No target answer answer is generated, only an instruction. A target answer can then be generated by other means, e.g. by using an LLM to solve the generated problem directly.

You are a **reasoning question generator assistant**. Your goal is to create a novel, and challenging reasoning question. You are provided the following seed questions:

Seed Question 1: **{INSTRUCTION 1}**
Seed Question 2: **{INSTRUCTION 2}**

Your task is to write a brand-new, self-contained reasoning question that meets the following requirements:
1. The question draws inspiration from the seed question without copying it verbatim, remaining novel and of comparable difficulty.
2. The question's final answer should be a single, unambiguous scalar value (e.g., an integer, reduced fraction, exact radical), or another answer type that can be verified in one step (e.g., 'yes/no,' a choice from A to D).
3. Do not include any solution, hint, or answer-—only the question statement itself.

Please put your generated problem strictly in the format of
[New Question Begin]{your_generated_question}[New Question End]

Figure 7: Self-Instruct instruction & target generation template (standard, without CoT) for verifiable reasoning tasks. This template prompts the LLM to generate a new reasoning question, followed by its corresponding target answer, without any reasoning.

You are a **reasoning question generator assistant**. Your goal is to create a novel, and challenging reasoning question. You are provided the following seed questions:

Seed Question 1: **{INSTRUCTION 1}**
Seed Question 2: **{INSTRUCTION 2}**

Your task is to:
1. Write a brand-new, self-contained reasoning question that meets the following requirements:
(a) The question draws inspiration from the seed question without copying it verbatim, remaining novel and of comparable difficulty.
(b) The question's final answer should be a single, unambiguous scalar value (e.g., an integer, reduced fraction, exact radical), or another answer type that can be verified in one step (e.g., 'yes/no,' a choice from A to D).
2. Then solve the new question and format your output as follows:
[New Question Begin]{your_generated_question}[New Question End]
[Final Answer to New Question Begin]\boxed{your_final_answer}[Final Answer to New Question End]

Figure 8: CoT-Self-Instruct (No-Solve) instruction generation template for verifiable reasoning tasks without answering (i.e., generate a question only).

> You are a **reasoning question generator assistant**. Your goal is to create a novel, and challenging reasoning question. You are provided the following seed questions:
>
> Seed Question 1: **{INSTRUCTION 1}**
> Seed Question 2: **{INSTRUCTION 2}**
>
> Your task is to write a brand-new, self-contained reasoning question that meets the following requirements:
> 1. The question draws inspiration from the seed question without copying it verbatim, remaining novel and of comparable difficulty.
> 2. The question's final answer should be a single, unambiguous scalar value (e.g., an integer, reduced fraction, exact radical), or another answer type that can be verified in one step (e.g., 'yes/no,' a choice from A to D).
> 3. Do not include any solution, hint, or answer-—only the question statement itself.
>
> Please reason step by step and put your generated problem strictly in the format of
> [New Question Begin]{your_generated_question}[New Question End]

Figure 9: Self-Instruct-Then-Solve (i.e. No CoT) instruction generation template for verifiable reasoning tasks.

> You are a **reasoning question generator assistant**. Your goal is to create a novel, and challenging reasoning question. You are provided the following seed questions:
>
> Seed Question 1: **{INSTRUCTION 1}**
> Seed Question 2: **{INSTRUCTION 2}**
>
> Your task is to:
> 1. Write a brand-new, self-contained reasoning question that meets the following requirements:
> (a) The question draws inspiration from the seed question without copying it verbatim, remaining novel and of comparable difficulty.
> (b) The question's final answer should be a single, unambiguous scalar value (e.g., an integer, reduced fraction, exact radical), or another answer type that can be verified in one step (e.g., 'yes/no,' a choice from A to D).
> 2. Then solve the new question and format your output as follows:
> [New Question Begin]{your_generated_question}[New Question End]
> [Final Answer to New Question Begin]\boxed{your_final_answer}[Final Answer to New Question End]

Table 4: **Results of CoT-Self-Instruct, comparing to baselines, for reasoning tasks on targets sampled from *Qwen3-4B*.** We conduct GRPO-training using Qwen3-4B-Base model on synthetic instructions generated by different templates, with targets sampled from Qwen3-4B. We report pass@1 averaged over 16 seeds. Two filter thresholds are used: SC = Self-Consistency Rate (i.e. the ratio majority votes over total votes) and RSc = RIP score (i.e. the quantile of minimum response score.)

| | # Train | Filter Thres. | MATH 500 | AIME 24 | AMC 23 | GPQA Diamond | Avg. |
|---|---|---|---|---|---|---|---|
| Self-Instruct | 5000 | - | 81.1 | 16.2 | 58.1 | 42.5 | 49.5 |
| + Self-Consistency Filter | 3467 | SC $\geq$ 0.5 | 83.6 | 18.5 | 68.5 | 44.1 | 53.6 |
| + RIP Filter | 2254 | RSc $\geq$ 0.5 | 84.5 | 21.2 | 65.9 | 45.5 | 54.5 |
| Self-Instruct-Then-Solve (NoCoT) | 5000 | - | 74.5 | 9.8 | 47.7 | 39.0 | 42.7 |
| + Answer-Consistency Filter | 646 | - | 75.6 | 12.9 | 53.9 | 38.1 | 45.1 |
| + Self-Consistency Filter | 3369 | SC $\geq$ 0.5 | 74.8 | 10.8 | 49.8 | 37.5 | 43.2 |
| + RIP Filter | 2162 | RSc $\geq$ 0.5 | 75.0 | 11.0 | 52.3 | 38.0 | 44.1 |
| CoT-Self-Instruct (NoSolve) | 5000 | - | 84.3 | 20.2 | 65.5 | 43.7 | 53.4 |
| + Self-Consistency Filter | 3972 | SC $\geq$ 0.5 | 84.7 | 24.8 | 67.5 | 44.9 | 55.5 |
| + RIP Filter | 2431 | RSc $\geq$ 0.5 | 84.9 | 24.2 | 72.3 | 44.6 | 56.5 |
| CoT-Self-Instruct | 5000 | - | 84.9 | 20.4 | 62.2 | 44.4 | 53.0 |
| + Answer-Consistency Filter | 2926 | - | **86.5** | **24.6** | **72.3** | **45.5** | **57.2** |
| + Self-Consistency filter | 4034 | SC $\geq$ 0.5 | 85.2 | 22.5 | 67.8 | 44.9 | 55.1 |
| + RIP filter | 2491 | RSc $\geq$ 0.5 | 85.7 | 24.4 | 70.5 | 44.4 | 56.2 |

Table 5: **893-train-size-matching results of CoT-Self-Instruct, comparing to baselines, for reasoning tasks on targets sampled from *Qwen3-4B*:** We conduct GRPO-training using Qwen3-4B-Base on selected s1k verifiable instructions and 893 synthetic instructions generated by different templates, with targets sampled from Qwen3-4B. We report pass@1 averaged over 16 seeds on MATH500, AMC23, AMIE24, GPQA-Diamond. Two filter thresholds are used: SC = Self-Consistency Rate (i.e. the ratio majority votes over total votes) and RSc = RIP score (i.e. the quantile of minimum response score.)

| | # Train | Filter Thres. | MATH 500 | AIME 24 | AMC 23 | GPQA Diamond | Avg. |
|---|---|---|---|---|---|---|---|
| ***s1k Instructions*** | | | | | | | |
| + Qwen3-4B Target | 893 | - | 71.3 | 13.7 | 51.5 | 38.7 | 43.8 |
| Self-Instruct | 893 | - | 80.5 | 17.2 | 57.3 | 41.3 | 49.1 |
| + Self-Consistency Filter | 893 | SC $\geq$ 0.5 | 81.9 | 20.0 | 62.8 | 41.5 | 51.5 |
| + RIP Filter | 893 | RSc $\geq$ 0.5 | 82.7 | 21.5 | 61.4 | 43.1 | 52.2 |
| CoT-Self-Instruct (NoSolve) | 893 | - | 82.5 | 20.2 | 61.7 | 41.4 | 51.4 |
| + Self-Consistency Filter | 893 | SC $\geq$ 0.5 | 83.6 | 20.6 | 61.7 | 43.0 | 52.2 |
| + RIP Filter | 893 | RSc $\geq$ 0.5 | 83.4 | **24.8** | 64.1 | 42.8 | 53.8 |
| CoT-Self-Instruct | 893 | - | 82.4 | 19.8 | 60.0 | 41.3 | 50.9 |
| + Answer-Consistency Filter | 893 | - | **83.7** | 23.1 | **66.1** | **44.1** | **54.2** |
| + RIP Filter | 893 | RSc $\geq$ 0.5 | 83.2 | 22.7 | 65.1 | 41.6 | 53.1 |
| + Self-Consistency Filter | 893 | SC $\geq$ 0.5 | 83.0 | 21.0 | 63.9 | 42.9 | 52.7 |

Table 6: **Results of CoT-Self-Instruct, comparing to baselines, for reasoning tasks on majority-voted targets sampled from *Qwen3-4B* model:** We conduct GRPO-training using Qwen3-4B-Base on synthetic instructions generated by different templates, with majority-voted targets sampled from Qwen3-4B. We report pass@1 averaged over 16 seeds. Different from Table 4 we use majority voted answers by Qwen3-4B model instead of single sampled responses. The conclusions are similar to Table 4.

| Majority-Voted Qwen3-4B Target | # Train | MATH 500 | AIME 24 | AMC 23 | GPQA Diamond | Avg. |
|---|---|---|---|---|---|---|
| Self-Instruct | 5000 | 80.8 | 15.6 | 57.2 | 43.7 | 49.3 |
| + Self-Consistency Filter | 3467 | 80.9 | 17.7 | 63.9 | **46.3** | 52.2 |
| CoT-Self-Instruct (NoSolve) | 5000 | 82.9 | **21.9** | 65.3 | 44.4 | 53.6 |
| + Self-Consistency Filter | 3972 | **83.7** | 21.3 | **68.8** | 44.2 | **54.5** |

Table 7: **Results of CoT-Self-Instruct, comparing to baselines, for reasoning tasks on Best-of-K targets sampled from _Qwen3-4B_ model** using the reward model infly/INF-ORM-Llama3.1-70B (Minghao Yang, 2024): We conduct GRPO-training using Qwen3-4B-Base on selected s1k verifiable instructions and synthetic instructions generated by different templates with targets sampled from Qwen3-4B. We report pass@1 averaged over 16 seeds on MATH500, AMC23, AMIE24, GPQA-Diamond.

| Best-of-K Qwen3-4B Targets | # Train | MATH 500 | AIME 24 | AMC 23 | GPQA Diamond | Avg. |
|---|---|---|---|---|---|---|
| Self-Instruct | 5000 | 83.8 | 18.8 | 62.0 | 44.4 | 52.2 |
| + RIP Filter | 2254 | 84.1 | 20.8 | 68.4 | 46.6 | 55.0 |
| CoT-Self-Instruct (NoSolve) | 5000 | 82.9 | 22.5 | 64.8 | 42.7 | 53.2 |
| + RIP Filter | 3651 | **85.2** | **24.4** | **71.1** | **46.8** | **56.9** |

Table 8: **Results of CoT-Self-Instruct, comparing to baselines, for reasoning tasks on targets sampled from _Qwen3-4B-Base_ model responses**: We conduct GRPO-training using Qwen3-4B-Base and report pass@1 averaged over 16 seeds on 4 benchmarks. Different from Table 4 we use answers sampled from the Qwen3-4B-Base model.

| | #Train | MATH 500 | AIME 24 | AMC 23 | GPQA Diamond | Avg. |
|---|---|---|---|---|---|---|
| Self-Instruct (Qwen3-4B-Base NoCoT) | 5000 | 75.7 | 13.1 | 51.4 | 28.0 | 42.1 |
| + Self-Consistency Filter | 2815 | 75.9 | 11.5 | 54.8 | 29.5 | 42.9 |
| + RIP Filter | 3492 | 75.4 | 12.5 | 51.2 | 28.2 | 41.8 |
| Self-Instruct (Qwen3-4B NoThink) | 5000 | 75.3 | 11.0 | **55.4** | 27.1 | 42.2 |
| + Self-Consistency Filter | 1757 | 75.1 | 11.9 | 52.2 | 27.0 | 41.5 |
| + RIP Filter | 2263 | 75.8 | 13.8 | 51.1 | 30.6 | 42.8 |
| CoT-Self-Instruct (Qwen3-4B NoSolve) | 5000 | 75.5 | 11.0 | 52.2 | 31.4 | 42.5 |
| + Self-Consistency Filter | 1672 | **77.0** | **15.4** | 50.5 | **35.4** | **44.6** |
| + RIP Filter | 2456 | 76.2 | 14.6 | 53.3 | 30.4 | 43.6 |

Table 9: **893-train-size-matching results of CoT-Self-Instruct, comparing to baselines, for reasoning tasks on targets sampled from _Qwen3-4B-Base_ model responses**. These experiments are the train-size-matching variants of Table 8.

| | # Train | MATH 500 | AIME 24 | AMC 23 | GPQA Diamond | Avg. |
|---|---|---|---|---|---|---|
| Qwen3-4B-Base (Zero-Shot) | - | 67.4 | 10.6 | 42.0 | 24.2 | 36.1 |
| s1k Prmpt + Qwen3-4B-Base Label | 893 | 75.1 | 10.4 | 47.3 | 28.7 | 40.4 |
| Self-Instruct (Qwen3-4B-Base NoCoT) | 893 | 75.3 | 10.0 | 51.7 | 27.1 | 41.0 |
| + Self-Consistency Filter | 893 | 75.7 | 11.7 | 51.3 | 28.2 | 41.7 |
| + RIP Filter | 893 | 76.2 | 12.5 | 50.5 | 29.2 | 42.1 |
| Self-Instruct (Qwen3-4B NoThink) | 893 | 75.3 | 10.0 | 51.7 | 27.1 | 41.0 |
| + Self-Consistency Filter | 893 | 76.2 | 10.2 | 53.3 | 26.6 | 41.6 |
| + RIP Filter | 893 | 76.0 | 11.9 | 52.2 | 31.3 | 42.8 |
| CoT-Self-Instruct (Qwen3-4B NoSolve) | 893 | 75.9 | 10.2 | 51.6 | 30.1 | 41.9 |
| + Self-Consistency Filter | 893 | 76.2 | 11.5 | **54.1** | **34.0** | **43.9** |
| + RIP Filter | 893 | **77.1** | **13.1** | 50.0 | 33.9 | 43.5 |

Table 10: **Results of CoT-Self-Instruct and other prompt templates for reasoning tasks on majority-voted targets from *Qwen3-4B-Base* model**: We conduct GRPO-training using Qwen3-4B-Base and report pass@1 averaged over 16 seeds on 4 benchmarks. Different from Table 8 we use majority-voted targets sampled from the Qwen3-4B-Base model.

| Majority-Voted Qwen3-4B-Base Target | # Train | MATH 500 | AIME 24 | AMC 23 | GPQA Diamond | Avg. |
|---|---|---|---|---|---|---|
| Self-Instruct (Qwen3-4B-Base) | 5000 | 76.2 | 11.7 | 51.7 | 30.5 | 42.5 |
| + Self-Consistency Filter | 2815 | **77.5** | 13.1 | 54.5 | 29.0 | 43.6 |
| CoT-Self-Instruct (Qwen3-4B-Base, No Solve) | 5000 | 76.3 | 13.1 | 49.7 | 30.2 | 42.3 |
| CoT-Self-Instruct (Qwen3-4B NoSolve) | 5000 | 76.1 | 12.3 | 54.5 | 31.3 | 43.5 |
| + Self-Consistency Filter | 1672 | 77.0 | **13.5** | **55.3** | **31.4** | **44.3** |

Table 11: **Additional comparisons for general instruction following tasks using different synthetic generation prompts**. CoT-Self-Instruct with long CoT generates synthetic data that outperforms short CoT and standard Self-Instruct templates. Both AlpacaEval 2 and ArenaHard are evaluated with two kinds of judge: GPT-4 Turbo and GPT-4o, with similar conclusions.

| | Training Method | AlpacaEval LC Winrate | | ArenaHard Score | | Avg. |
|---|---|---|---|---|---|---|
| | | GPT-4 Turbo | GPT-4o | GPT-4 Turbo | GPT-4o | |
| **Self-Instruct** (No CoT) | DPO | 52.9 | 46.0 | 51.8 | 39.2 | 47.4 |
| + RIP Filter | DPO | 55.2 | 46.1 | 55.6 | 39.5 | 49.1 |
| **CoT-Self-Instruct** (Short CoT) | DPO | 56.5 | 44.3 | 51.6 | 34.1 | 46.6 |
| + RIP Filter | DPO | 59.0 | 37.7 | 54.3 | 37.5 | 47.1 |
| **CoT-Self-Instruct** | DPO | 58.5 | 48.6 | 62.0 | 46.7 | 53.9 |
| + RIP Filter | DPO | 63.2 | 49.4 | 60.2 | 45.8 | 54.7 |

## 8 THE USE OF LARGE LANGUAGE MODELS(LLM)

In our project, we use LLM for writing polishing.

