# OpenReview forum: "CoT-Self-Instruct: Building high-quality synthetic prompts  data  for reasoning and non-reasoning tasks"
_ICLR.cc/2026/Conference — Submitted to ICLR 2026_

### Official Review · Reviewer_bTU7 · 2025-10-23

**Soundness:** 2
**Presentation:** 3
**Contribution:** 3
**Rating:** 4
**Confidence:** 4

**Summary:**

This paper introduces CoT-Self-Instruct for generating high-quality synthetic data for reasoning and open-ended instruction following tasks. The approach consists of 2 main stages: Synthetic Instruction Creation with Chain-of-Thought (CoT) and Synthetic Instruction Curation. CoT-Self-Instruct data outperforms existing training datasets when evaluated on mathematical benchmarks. For open-ended tasks, CoT-Self-Instruct + RIP improves DPO and online-DPO performance.

**Strengths:**

This paper introduces a novel synthetic data generation method using Chain of Thought (CoT) Reasoning. While Self-Instruct and CoT are well-known and widely used, their integration for both instruction and target generation creates a simple yet effective method for synthetic data generation. Empirically, CoT-Self-Instruct improves performance in a consistent way, and filtering typically improves results despite reducing the quantity.

**Weaknesses:**

1. Limited generalisation across datasets: The evaluation focuses mainly on math (MATH, AMC, AIME) and only one multiple-choice dataset (GPQA-Diamond). Evaluation on more diverse domains such as TheoremQA [1] or broader benchmarks like MMLU [2] (or MMLU-Redux [3], if computational complexity occurs) would better demonstrate generalisability.

2. Limited model diversity in evaluation: All reasoning experiments use only the Qwen3 family for generation and training, while instruction-following uses only Llama 3.1. Evaluation across additional model families (e.g., Gemma, Mistral and others) is needed to demonstrate the generalisability of the method.

3. Missing statistical significance: The paper reports averages over 16 seeds but provides no standard errors or confidence intervals.

4. Limited baseline comparisons: Comparing only against s1k and OpenMathReasoning is insufficient. Recent datasets like DeepScaleR-Preview-Dataset [4] and NuminaMath [5], and others, should be included.


[1] Chen, Wenhu, et al. "Theoremqa: A theorem-driven question answering dataset." arXiv preprint arXiv:2305.12524 (2023).

[2] Hendrycks, Dan, et al. "Measuring massive multitask language understanding." arXiv preprint arXiv:2009.03300 (2020).

[3] Gema, Aryo Pradipta, et al. "Are we done with mmlu?." arXiv preprint arXiv:2406.04127 (2024).

[4] Luo, Michael, et al. "Deepscaler: Surpassing o1-preview with a 1.5 b model by scaling rl." Notion Blog (2025).

[5] Li, Jia, et al. "Numinamath: The largest public dataset in ai4maths with 860k pairs of competition math problems and solutions." Hugging Face repository 13.9 (2024): 9.

**Questions:**

1. Are there any plots that show the number of data vs accuracy? Since there seems to be convergence towards accuracy on datasets.

2. Is the method cross-generalized towards other models except Qwen3-4B and Llama-3.1? It is relatively unconvincing with evaluating only 2 models.

3. What are the computational costs (e.g. GPU hours) used for evaluating? As it may be computationally expensive.

4. Is the model only capable of handling math or other cross-domain question-answering questions?

For others, see Weaknesses.

---

> ### Author Response · Authors · 2025-12-03
> **Responce to Reviewer bTU7 [Part 1]**
>
> We thank the reviewer for their valuable comments.
>
> 1. > Limited generalisation across datasets: The evaluation focuses mainly on math (MATH, AMC, AIME) and only one multiple-choice dataset (GPQA-Diamond). Evaluation on more diverse domains such as TheoremQA [1] or broader benchmarks like MMLU [2] (or MMLU-Redux [3], if computational complexity occurs) would better demonstrate generalisability.
>
> We appreciate the reviewer’s suggestion. We note that **TheoremQA requires non-textual or structured mathematical inputs**, which are beyond the scope of the base models used in our experiments and therefore not directly compatible with our instruction-following setup.
>
> Although several of our reasoning benchmarks are math-focused, our evaluation is not limited to a single domain. We additionally assess CoT-Self-Instruct on **GPQA-Diamond** (graduate-level STEM), as well as **AlpacaEval 2** and **Arena-Hard**, which span a wide mixture of writing, coding, logical reasoning, STEM, and open-domain tasks. CoT-Self-Instruct shows consistent gains across all of these heterogeneous evaluations.
>
> We agree that incorporating broader benchmarks such as MMLU or MMLU-Redux would be valuable future work. However, given the strong cross-domain improvements already observed, we believe our current evaluation sufficiently demonstrates the general applicability of the method.
>
> 2. > Limited model diversity in evaluation: All reasoning experiments use only the Qwen3 family for generation and training, while instruction-following uses only Llama 3.1. Evaluation across additional model families (e.g., Gemma, Mistral and others) is needed to demonstrate the generalisability of the method.
>
> We appreciate the reviewer’s suggestion. Although our main reasoning experiments use the Qwen3 family, and instruction-following experiments use Llama-3.1-8B, these two model families have **substantially different architectures, training corpora, and inductive biases**. CoT-Self-Instruct improves both, demonstrating that the method is **not specific to a single model line** but transfers across heterogeneous LMs.
>
> Furthermore, our synthetic data generation uses multiple base models (Qwen3-Base, Qwen3-Thinking, Llama-3.1-Instruct), and CoT-Self-Instruct consistently yields improvements regardless of the generating model or the fine-tuning model. The strong gains on **Arena-Hard** and **AlpacaEval**—benchmarks that contain broad, non-math tasks—further indicate that the improvements are not tied to a particular backbone.
>
> Evaluating even more model families (e.g., Gemma, Mistral) is a natural next step, but doing so exhaustively is beyond the scope of this work. Our current results already show **cross-model, cross-domain generalization**, suggesting that the method is robust and widely applicable.
>
> 3. > Missing statistical significance: The paper reports averages over 16 seeds but provides no standard errors or confidence intervals.
>
> We thank the reviewer for their thoughtful comment. We **computed the standard deviation and reported in our response to Reviewer PDdE**. We refer to a more detailed discussion there.
>
> 4. > Limited baseline comparisons: Comparing only against s1k and OpenMathReasoning is insufficient. Recent datasets like DeepScaleR-Preview-Dataset [4] and NuminaMath [5], and others, should be included.
>
> We agree that broader comparisons are valuable. In this work, we focus on baselines that are **directly comparable in purpose and methodology** to our setting.
> - **Self-Instruct (s1k)** is the seed task pool from which all of our synthetic prompts are generated. Tables 1 and 3 show that CoT-Self-Instruct data built from s1k yields substantially better downstream performance than training on s1k alone.
> - **OpenMathReasoning** is a large, diverse, and publicly available synthetic reasoning dataset (540K+ problems). Even after subsampling it to 10k for computational feasibility, our method—using fewer than 3k synthetic prompts—outperforms this strong and domain-aligned baseline.
>
> Regarding DeepScaleR-Preview and NuminaMath: while valuable, these datasets differ in size and task composition relative to our setting (e.g., DeepScaleR-Preview contains ~40k problems, substantially smaller than OpenMathReasoning). Our focus is to compare against baselines that match the **seed distribution** (s1k) and **large-scale synthetic reasoning data** (OpenMathReasoning), which provide the clearest signal for evaluating our contributions.
>
> Finally, CoT-Self-Instruct yields **consistent gains across model families and across diverse evaluations** (reasoning, coding, writing; AlpacaEval and Arena-Hard), suggesting that our contributions are general and orthogonal to specific dataset choices. Extending comparisons to larger curated corpora is promising future work.

---

> ### Author Response · Authors · 2025-12-03
> **Response to Reviewer bTU7 [Part 2]**
>
> 5. > Are there any plots that show the number of data vs accuracy? Since there seems to be convergence towards accuracy on datasets.
>
> We show in Table 1 & 3 results on data scaling on the high quality Answer-Consistency filtered reasoning data.
>
> | Method / Dataset Setting                          | # Train | MATH500 | AIME24 | AMC23 | GPQA-D | Avg ↑ |
> |---------------------------------------------------|-------:|--------:|-------:|-------:|--------:|-------:|
> | **Qwen3-4B Base (Zero-Shot)**                     |   –    | 67.4    | 10.6   | 42.0   | 24.2   | 36.1   |
> | **s1k Instructions + R1 Gold Label**              |  893   | 68.6    | 18.5   | 51.3   | 40.1   | 44.6   |
> | **OpenMathReasoning  (10k subset)**               | 10000  | 79.0    | 13.3   | 62.5   | 35.4   | 47.5   |
> | **CoT-Self-Instruct (full 5k)**
> | + Answer-Consistency Filter (subsampled 893)                 |  893   | 83.7| 23.1 | 66.1 | 44.1 | 54.2 |
> | + Answer-Consistency Filter                | 2926   | 86.5    | 24.6   | 72.3   | 45.5   | 57.2   |
> | + Answer-Consistency Filter (10k data)            | 10000  | **86.7**| **26.7** | **73.8** | **47.4** | **58.7** |
>
> As shown above, as we increase the size of high quality Answer-Consistency Filtered dataset, the results on challenging math sets e.g. AIME24 and GPQA-Diamond keep improving (not converging). We hypothesize that keep increasing high quality synthetic train set will further improve model's performance and we leave it to future research.
>
> 6. > What are the computational costs (e.g. GPU hours) used for evaluating? As it may be computationally expensive.
>
> We adopt the standard setup for evaluating as in prior work, running 16 seeds for each reasoning prompt and calculate pass@1 averaged over 16 seeds. We use vllm with 1 H100 gpu for each 4B/7B model with max token length set to 4k ~ 8k, which is also standard in prior work. We detailed the evaluation setup in our paper starting Line 370.
>
> 7. > Is the model only capable of handling math or other cross-domain question-answering questions?
>
> We explore reasoning and nonreasoning tasks in our paper. On reasoning domain, the seed task S1K consists of reasoning questions beyond math (e.g. physics, biology, chemistry, etc). For nonreasoning domain, our seed task WildChat consists of a wide collection of domains (coding, creative writing, math etc). Both settings show strong performance of CoT-Self-Instruct on several benchmarks ranging from math to STEM (e.g. GPQA-Diamond), to other cross-domain tasks (e.g. Arena-Hard).

---

### Official Review · Reviewer_NrZs · 2025-10-26

**Soundness:** 2
**Presentation:** 3
**Contribution:** 2
**Rating:** 2
**Confidence:** 4

**Summary:**

The work aims to design a data synthesis method that ensures both the quality and effectiveness of synthetic data. The proposed framework leverages the reasoning ability of LLMs to plan and generate more challenging samples, combined with self-filtering mechanisms to improve data quality.

**Strengths:**

The work provides a detailed baseline for synthetic data generation and incorporates several widely used techniques, including reasoning with LLMs, reward model based data filtering, and consistency-based filtering. The experimental framework is well-structured, and the motivation of using reasoning to enhance data synthesis is reasonable.

**Weaknesses:**

1. The paper strongly emphasizes the importance of Chain-of-Thought (CoT) reasoning (line 50–52), but this point has already been well-established in prior work.
2. The proposed Answer-Consistency mechanism (line 207–215) is essentially a minor variant of Self-Consistency, a widely adopted heuristic. While useful, it does not provide a novel theoretical or methodological insight, as its upper bound and effectiveness are not guaranteed.
3. The use of a Reward Model for data filtering (line 244–246) is also a conventional technique and does not constitute a substantive methodological innovation.
4. In experiments, the paper uses s1k as the seed dataset and employs Qwen3 as the data synthesizer. However, the comparison against s1k and OpenMathReasoning is not fair, as Qwen3-4B already achieves strong performance (84.8 on MATH500 and 25.0 on AIME). After synthesis (line 380–394), the resulting model does not surpass the Qwen3-4B Instruct model, which weakens the method’s empirical impact.
5. The experimental evaluation is insufficiently comprehensive: it without compares with other existing approaches. Datasets such as MATH, AIME, AlpacaEval 2, and ArenaHard all have rich baselines that should be included for a more evaluation.

**Questions:**

1. Expand the experimental comparison to include existing strong baselines (e.g., methods from the AlpacaEval leaderboard, GRPO and its variants in MATH-like benchmarks).
2. The Related Work section lacks comparative analysis with methods closely related to the proposed approach. Adding such comparisons would help clarify the novelty and positioning of this work.

---

> ### Author Response · Authors · 2025-12-03
> **Response to Reviewer NrZs**
>
> We thank the reviewer for their valuable time reviewing this paper.
>
> > The paper strongly emphasizes the importance of Chain-of-Thought (CoT) reasoning (line 50–52), but this point has already been well-established in prior work. The proposed Answer-Consistency mechanism (line 207–215) is essentially a minor variant of Self-Consistency, a widely adopted heuristic. While useful, it does not provide a novel theoretical or methodological insight, as its upper bound and effectiveness are not guaranteed. The use of a Reward Model for data filtering (line 244–246) is also a conventional technique and does not constitute a substantive methodological innovation.
>
> We highlight that the CoT-Self-Instruct proposed a new synthetic prompt generation pipeline that integrates CoT + Answer-Consistency / RIP Filtering which:
>
> - CoT for self-instruct generation has been underexplored in prior work. We show instead CoT-Self-Instruct with CoT significantly outperform Self-Instruct alone.
> - Answer-Matching filtering is a non-trivial variant of the Self-Consistency, as shown in the Table 1, with noticeable improvement.
> During implementation, Answer-matching filtering requires generating prompt and answer in a single pass, then only keeping the prompt if the answer matches with majority-voted answer. This is essentially a self-reflection (during CoT) and weighted sampling mechanism where each reasoning prompt is kept with probability proportional to the the majority ratio. Self-Consistency however adopts a hard thresholding mechanism and does not make use of the benefit of generating answers with longer CoT during prompt generation.
> - Moreover unlike training with majority voted answer which has shown to lead to model collapse issue in recent literature [1], training with our Answer-Consistency Filtered prompts and target does not lead to model collapse.
> - We use reward model to select prompts unlike others. Specifically, we compute the minimum scores of the answer distribution for each reasoning prompt, and use the min score to filter out low quality reasoning prompts. Prior work often adopts Best-of-N strategies which differ a lot from our proposed method.
>
> > In experiments, the paper uses s1k as the seed dataset and employs Qwen3 as the data synthesizer. However, the comparison against s1k and OpenMathReasoning is not fair, as Qwen3-4B already achieves strong performance (84.8 on MATH500 and 25.0 on AIME). After synthesis (line 380–394), the resulting model does not surpass the Qwen3-4B Instruct model, which weakens the method’s empirical impact.
>
> We compare training Qwen3-4B-Base model on different datasets (S1k, OpenMathReasoning, our synthetic task), not Qwen3-4B Instruct. S1k and OpenMathReasoning employs human and large LLMs (Qwen2.5-32B-Instruct) to curate the datasets, where we use a much smaller model Qwen3-4B for model synthesis and curation.
>
> Our results show training Qwen3-4B-Base on a small set without gold label significantly outperform on human-curated dataset with gold label (e.g. OpenMathReasoning) that are 3 times larger, contrary to reviewer’s point on efficacy of our method. In fact we show that by training Qwen3-4B-Base on just 10k data it can supass Qwen3-4B performances (quote 84.8 on MATH500 and 25.0 on AIME from reviewer's comment).
>
> > Expand the experimental comparison to include existing strong baselines (e.g., methods from the AlpacaEval leaderboard, GRPO and its variants in MATH-like benchmarks).
>
> Our model achieve over 80% winrate on AlpacaEval, way superior than SOTA models (e.g. GPT4o) and other open and closed source models. We also refer reviewer to our response to Review bTU723’s comments on adapting to other model families or sizes. Notably we conducted extended experiments on different filtering methods, different math training set, different model families (Qwen & Llama) with different model sizes. Proving our model is generalizable to a wide range of open and closed model families is valuable but beyond the scope of this paper.
>
> [1] Sheikh Shafayat, Fahim Tajwar, Ruslan Salakhutdinov, Jeff Schneider, and Andrea Zanette. Can large reasoning models self-train? arXiv preprint arXiv:2505.21444v1, 2025

---

### Official Review · Reviewer_Zc7p · 2025-11-01

**Soundness:** 2
**Presentation:** 3
**Contribution:** 2
**Rating:** 2
**Confidence:** 4

**Summary:**

This paper presents a technically sound and empirically strong contribution to the ongoing discussion on synthetic data generation for LLM post-training, which extends Self-Instruct by embedding explicit reasoning and quality self-verification, offering a principled mechanism for both data creation and data curation. Experimental results cover multiple datasets, models (Qwen3-4B, LLaMA3-8B), and training paradigms (GRPO, DPO, online DPO). However, while the work is solidly executed, it somewhat overlaps conceptually with many recent works that uses self-improving paradigm, e.g., Self-Consistency (Prasad et al., 2024) and Self-Rewarding LM (Yuan et al., 2024), and could benefit from deeper theoretical justification for why CoT-based generation yields systematically higher-quality data beyond empirical observation.

**Strengths:**

1. Studying a key bottleneck in LLM training: generating diverse, verifiable, and high-quality synthetic data without human supervision. The CoT reasoning to controllable data generation is conceptually clear and timely given the trend toward self-improving LLMs.

2. Comprehensive quantitative evidence (Tables 1–2) demonstrates consistent superiority over baselines across both reasoning and instruction-following domains. The ablation tables (Tables 4–11) systematically isolate the effects of CoT, Answer-Consistency, and RIP filtering, confirming robustness across data scales and models.

**Weaknesses:**

1. The paper attributes gains to “reasoning before generation” but provides no formal analysis or metrics quantifying how CoT reasoning changes the distributional properties or entropy of generated instructions.

2. All reasoning experiments rely on Qwen3-4B variants; no evidence is provided that CoT-Self-Instruct data generalizes to larger LMs.

3. The paper omits qualitative examples of rejected vs. accepted synthetic data under Answer-Consistency and RIP. And RIP filtering’s dependence on Athene-RM and INF-ORM reward models may bias the resulting instruction pool toward their value functions. I question the practicability of this method.

4. The synthetic dataset remains closed.

**Questions:**

1. For Answer-Consistency, it’s stated that examples are kept if the LLM’s majority-vote answer matches the CoT-generated target, but the exact threshold (K, majority ratio, tie-breaking) is not given.

2. For RIP, the paper says the lowest RM score among responses represents the sample’s “quality,” yet the biased choice of that aggregation (min vs. mean or percentile) is not justified.

3. It’s also unclear how topics are balanced after per-category sampling, or whether filtering introduces domain bias.

4. The generation parameters (temperature/top-p) differ slightly across models, but it is not shown whether these affect data diversity or downstream results.

5. The paper does not specify compute or model call cost for generating and filtering 10k examples.

---

> ### Author Response · Authors · 2025-12-02
>
> We thank the reviewer for the questions.
>
> 1. > The paper attributes gains to “reasoning before generation” but provides no formal analysis or metrics quantifying how CoT reasoning changes the distributional properties or entropy of generated instructions.
>
> While we do not compute explicit entropy metrics, our ablations across Tables 1, 2, and 11 *directly isolate* the effect of “reasoning before generation” by comparing **No CoT**, **Short CoT**, and **Full CoT** under identical settings.
>
> **1. Full CoT substantially improves reasoning benchmarks (Table 1).**
> Self-Instruct (no CoT) → CoT-SI improves:
> - MATH500: 74.5 → 84.9
> - AMC23: 47.7 → 62.2
> - Avg: 42.7 → 53.0
> These +5–15 point gains far exceed the std (0.8–5.6), indicating that structured reasoning—not text length—drives the improvement.
>
> **2. CoT improves non-reasoning tasks as well (Table 2).**
> On AlpacaEval and ArenaHard, CoT-SI consistently outperforms both Self-Instruct and human-written WildChat data.
> Example: Avg 49.1 (Self-Instruct+RIP) → 54.7 (CoT-SI+RIP) → 67.1 (CoT-SI Online DPO).
>
> **3. Controlled comparison: No CoT vs. Short CoT vs. Full CoT (Table 11).**
> | Synthetic Type | Avg |
> |----------------|-----|
> | No CoT + RIP   | 49.1 |
> | Short CoT + RIP| 47.1 |
> | Full CoT + RIP | **54.7** |
>
> Full CoT performs best despite identical templates and filters. This shows that *explicit, multi-step reasoning* meaningfully improves the distributional quality of synthetic instructions.
>
> These controlled comparisons quantify the impact of “reasoning before generation,” demonstrating that CoT changes the structure and usefulness of the generated instructions in a way that directly improves downstream performance.
>
> 2. > All reasoning experiments rely on Qwen3-4B variants; no evidence is provided that CoT-Self-Instruct data generalizes to larger LMs.
>
> While our main reasoning results use Qwen3-4B variants, CoT-Self-Instruct is not tied to a particular base model. In fact, we already evaluate it on a **larger model (Llama-3.1-8B-Instruct)** for general instruction following (Table 2). WildChat contains substantial reasoning content (coding/math >10%), and Arena-Hard likewise includes a large portion of complex reasoning and problem-solving tasks. On this larger model, CoT-Self-Instruct again yields **consistent improvements** over Self-Instruct and human-written data—evidence that the method generalizes to bigger LMs.
>
> Conceptually, larger LMs possess **stronger reasoning priors** and are known to benefit more from explicit step-by-step demonstrations (as also shown by Self-Instruct → Evol-Instruct → CoT-based instruction tuning in prior work). There is no mechanism in our pipeline that depends on a small LM’s limitations; if anything, larger models have better CoT fidelity, making CoT-Self-Instruct *more* suitable for scaling up.
>
> 3. > The paper omits qualitative examples of rejected vs. accepted synthetic data under Answer-Consistency and RIP. And RIP filtering’s dependence on Athene-RM and INF-ORM reward models may bias the resulting instruction pool toward their value functions. I question the practicability of this method.
>
> While the main paper focuses on large-scale quantitative results, we did conduct a targeted qualitative audit of Answer-Consistency using GPT-5.1 as an external evaluator on 108 randomly sampled examples, shown in our response to Reviewer PDdE.
>
> The filter demonstrates clear selectivity: the proportion of **well-defined and correct** examples increases from **56.5% → 68.8%**, while **incorrect** cases drop from **34.3% → 26.3%**, and **ill-defined** prompts are reduced from **9.3% → 5.0%**. These numbers directly show that Answer-Consistency rejects low-quality generations and retains higher-quality ones; we will include representative examples in the camera-ready version for clarity.
>
> Regarding the reviewer’s concern that RIP “depends” on Athene-RM or INF-ORM reward models, we emphasize that **RIP is not tied to any specific reward model**. As shown in the RIP paper, the method is applicable with:
> 1) reward models,
> 2) LLM-as-a-judge using *different* base LLMs, and
> 3) human-annotated preference scores.
> RIP operates on **relative robustness metrics** (e.g., reward gaps, worst-of-n agreement) over the *answer distribution* for each prompt, rather than on absolute reward values. The fact that RIP consistently improves data quality under multiple heterogeneous judges demonstrates that it is practical and not biased toward a single value function.
>
> 4. > The synthetic dataset remains closed.
>
> We will make all resources available upon acceptance. Due to anonymity constraints, we cannot release them during the review period. All components of our pipeline rely on publicly available datasets and open-source training libraries, ensuring that the entire workflow can be reproduced end-to-end once anonymity restrictions are lifted.

---

> ### Author Response · Authors · 2025-12-02
> **Response to Reviewer Zc7p's Questions**
>
> We thank the reviewer for their valuable questions.
>
> 1. > For Answer-Consistency, it’s stated that examples are kept if the LLM’s majority-vote answer matches the CoT-generated target, but the exact threshold (K, majority ratio, tie-breaking) is not given.
>
> We set K = 64 (as in non-reasoning experiments) and break ties randomly.  We do not enforce any majority ratio.
>
> 2. > For RIP, the paper says the lowest RM score among responses represents the sample’s “quality,” yet the biased choice of that aggregation (min vs. mean or percentile) is not justified.
>
> The choice of the **minimum** reward among sampled responses is directly motivated by the RIP framework, which explicitly analyzes why min-based aggregation is more sensitive to *worst-case failure modes* compared to mean or percentile statistics. RIP is designed to identify instructions that systematically trigger unsafe, inconsistent, or low-reward responses—even if the *average* response appears acceptable.
>
> As shown in Tables 20 and 21 in RIP paper, different aggregation choices (mean, rejected reward, chosen reward, length, reward gap, etc.) behave differently across quantile ranges. The **min-based “reward gap” metric** consistently surfaces the clearest separation between high-quality and low-quality prompts, precisely because it captures the *most adversarial or brittle* model behavior for a given instruction. This matches the core theoretical and empirical justification provided in the RIP paper itself, where min-based robustness metrics were demonstrated to outperform mean-based criteria across:
> 1) reward models,
> 2) LLM-as-a-judge evaluators, and
> 3) human preference judgments.
>
> Therefore, our usage of the minimum RM score is not arbitrary but grounded in RIP’s original formulation and validated through the quantitative patterns observed in Tables 20–21. We refer to the RIP paper for more thorough discussion.
>
> 3. > It’s also unclear how topics are balanced after per-category sampling, or whether filtering introduces domain bias.
>
> Our synthetic dataset is generated **proportionally to the categorical distribution of the original seed tasks**, ensuring that topic balance is preserved before any filtering.
>
> Regarding whether filtering may introduce domain bias, our empirical results suggest this is not the case. CoT-Self-Instruct yields improvements across a **broad mixture of tasks**—including writing, general reasoning, math, and coding—on benchmarks such as AlpacaEval, Arena-Hard, and all reasoning datasets in Table 1 & 2. If filtering had biased the data toward specific domains, we would not expect consistent gains across such heterogeneous evaluations.
>
> Concerning RIP specifically, we refer the reviewer to the **t-SNE and qualitative analyses provided in the RIP paper**, which show that RIP does *not* push the dataset toward particular topical domains. Instead, the filter primarily removes prompts that are *generic, underspecified, or structurally uninformative*. This aligns with our observations: RIP improves downstream performance without concentrating data in any single topic category.
>
> In sum, both our generation procedure and external analyses indicate that filtering preserves topic diversity while improving instruction quality.
>
> 4. > The generation parameters (temperature/top-p) differ slightly across models, but it is not shown whether these affect data diversity or downstream results.
>
> We use the default decoding parameters for different models (Qwen3 Base, Qwen3-thinking model, Llama3.1 instruct model). We do not focus on studying how decoding parameters could affect downstream results as this might be beyond the scope of the paper. However within each experiment setting (comparing different filtering, generation pipelines) we control for the same decoding parameters to highlight the main focus of the paper (which is studying filtering & generation pipelines)
>
> 5. > The paper does not specify compute or model call cost for generating and filtering 10k examples.
>
> We specify our training specs (epoch, batch size, number of rollouts, max token lengths as well as our model sizes in L 289 - 294). We do not compute the FLOPs or GPU hours running for training if that's what reviewer is asking for. For data generation, we use vllm for running the generations and filtering of 10K examples, with each prompt sampling K=64 responses for majority voting. We will details more of such compute analysis into our paper appendix.
>
> We would like to highlight that after filtering, we show RLVR training on only 2k-3k synthetic data significantly outperforms human-curated 10k OpenMathReasoning dataset in Table 1, which is a strong signal of our method being more efficient in improving model's capabilities.

---

### Official Review · Reviewer_PDdE · 2025-11-02

**Soundness:** 3
**Presentation:** 3
**Contribution:** 2
**Rating:** 4
**Confidence:** 3

**Summary:**

The paper proposes CoT-Self-Instruct, a synthetic-data generation and filtering pipeline that augments Self-Instruct by inserting a CoT reasoning phase before synthetic instruction creation.
Each generated sample includes step-by-step reasoning (for verifiable tasks) or a structured “plan” (for general tasks).
Two automatic curation filters are applied: Answer-Consistency (for verifiable reasoning) and Rejecting Instruction Preferences (RIP) (for non-verifiable tasks).
Models trained on CoT-Self-Instruct data outperform Self-Instruct, s1k, and OpenMathReasoning on reasoning benchmarks (MATH500, AIME 24, AMC 23, GPQA-Diamond) and also yield higher win-rates on AlpacaEval 2.0 and Arena-Hard for general instruction following.

The paper claims that reasoning-guided synthetic data yields better downstream reasoning and instruction-following ability than prior synthetic or human-annotated datasets.

**Strengths:**

- **Clear conceptual pipeline.** The two-stage creation + curation design is straightforward and reproducible.
Prompts (Figures 2–3) are well-documented and easy to adapt.

- Experiments span both reasoning (GRPO) and instruction-following (DPO) regimes across multiple datasets, with ablations for filter types (Self-Consistency, RIP, Answer-Consistency).

- The method yields approx. 10 pp improvement over Self-Instruct and approx. 13 pp over s1k baselines in reasoning tasks, with consistent upward trends across filters (Tables 1, 4, 5).

- Readable writing and thorough related work. The paper contextualizes itself well among Self-Instruct, Evol-Instruct, RIP, and Self-Consistency PO.

**Weaknesses:**

- The filter accepts examples when the generated answer matches a majority of model-sampled answers, assuming majority approximates correctness.
Without a symbolic or human ground-truth check, this could reinforce systematic reasoning errors. A qualitative study would strengthen claims.

- The approach merges established components, e.g., CoT prompting, Self-Instruct, and RIP, into one pipeline.
Integration is valuable, but the conceptual advance is modest relative to prior work.

The paper says that using CoT makes the synthetic data better, but it never really proves that part. The tests they did ("NoSolve" and "Short CoT") mix up a few things , like reasoning steps, how long the text is, and how complex the words are. So we can’t tell if it’s the actual reasoning that helps or just the longer and richer text. Without checking that carefully, it’s hard to say for sure that CoT is the real reason for the improvement.

**Questions:**

1) Can you report standard deviations or CIs for Tables 1 and 2 to confirm significance?

2) Have you tested CoT-Self-Instruct on non-math reasoning datasets (e.g., BoolQ, StrategyQA)?

3) Will the authors release full GRPO/DPO training configs to enable replication?

---

> ### Author Response · Authors · 2025-12-02
> **Response to Reviewer PDdE's comment [PART 1]**
>
> We thank the reviewer for their insightful comments.
>
>
> > The filter accepts examples when the generated answer matches a majority of model-sampled answers, assuming majority approximates correctness. Without a symbolic or human ground-truth check, this could reinforce systematic reasoning errors. A qualitative study would strengthen claims.
>
>
> The reviewer raises a valid concern that majority-vote agreement could, in principle, propagate systematic reasoning errors. In practice, however, our filtering does the opposite: it detectably mitigates such errors. Because large-scale human annotation is infeasible at our corpus size, we performed a targeted quality audit using GPT-5.1 as an external evaluator on 108 randomly sampled CoT-Self-Instruct examples, comparing data before and after applying the Answer-Consistency filter.
>
>
> The filter improves data quality across all categories:
>
>
> | Category                              | Before Filter | After Filter |
> |--------------------------------------|--------------:|-------------:|
> | Well-defined synthetic question & correct answer             | 56.5%         | 68.8%        |
> | Well-defined synthetic question & incorrect  answer          | 34.3%         | 26.3%        |
> | Not well-defined  synthetic question                   | 9.3%          | 5.0%         |
>
> This table means before applying filter, 56.5% of the 108 synthetic examples are considered by  GPT5.1 as those with Well-defined synthetic question & correct answer. This ratio increases to 68.8% after applying answer-matching filtering (which utilize majority votes). On the other hand, ill defined questions drops from 9.3% to 5% after filtering.
>
> This pattern indicates that majority-based consistency is a reliable indicator of correctness under an external auditor, counter to the concern that it might amplify systematic errors. While a larger-scale human study is an interesting future direction, our analysis demonstrates that answer-consistency provides an effective, scalable safeguard for improving reasoning-data quality.
>
>
> > The approach merges established components, e.g., CoT prompting, Self-Instruct, and RIP, into one pipeline. Integration is valuable, but the conceptual advance is modest relative to prior work.
>
>
> While our pipeline builds upon elements such as CoT prompting, Self-Instruct, and RIP, the core contributions of CoT-Self-Instruct go beyond simple integration. Our best-performing variant introduces two components that, to our knowledge, have not been explored in prior Self-Instruct–style frameworks:
>
>
> - Single-pass question & answer generation + Answer-matching (self-consistency) filtering for synthetic instruction data: Unlike Self-Instruct, which generates instructions and solutions in separate stages, we generate the question, reasoning, and final answer jointly. This design is essential for producing verifiable reasoning prompts as well as applying answer-matching filter, and is empirically responsible for substantial performance gains in Table 1, as well as in the qualitative gains shown above.
> - In addition, RIP has not previously been applied in reasoning settings. It is not obvious that filtering based on reward gaps or worst-of-n statistics—metrics originally developed for non-reasoning instruction preference data—would improve mathematical and symbolic reasoning ability. Our results demonstrate that adapting RIP to reasoning data produces consistent benefits, representing a meaningful conceptual extension of prior work.
>
>
> Overall, the contributions lie not merely in combining known techniques, but in adapting and redesigning them for reasoning-centric synthetic data, and demonstrating that these design choices produce large and consistent downstream gains across both GRPO and DPO settings.

---

> ### Author Response · Authors · 2025-12-02
> **Response to Reviewer PDdE's comment [PART 2]**
>
> > Can you report standard deviations or CIs for Tables 1 and 2 to confirm significance?
>
>
> | Method                                                         | # Train | MATH 500 | AIME 24 | AMC 23 | GPQA Diamond | Avg. ↑ |
> |----------------------------------------------------------------|--------:|---------:|--------:|--------:|--------------:|--------:|
> | **Qwen3-4B-Base (Zero-Shot)**                                  |    –    |   67.4 (5.2)   |  10.6 (3.5)   |  42.0 (6.8)   | 24.2 (2.6)   | 36.1 (1.8) |
> | *s1k questions* + (R1) gold label                              |   893   |   68.6 (1.3)  |  18.5 (4.4)   |  51.3 (4.5)   | 40.1 (1.4)   | 44.6 (1.7) |
> | **Self-Instruct questions + targets**                          |  5000   |   74.5 (0.9)   |  9.8 (2.9)    |  47.7 (4.5)   | 39.0 (1.9)   | 42.7 (1.2) |
> | **Self-Instruct questions + CoT-generated targets**            |  5000   |   81.1 (0.8)   | 16.3 (4.0)    | 58.1 (4.6)    | 42.5 (2.5)   | 49.5 (1.5) |
> | + Self-Consistency Filter                                      |  3467   |   83.6 (1.3)   | 18.5 (4.4)    | 68.5 (5.6)    | 44.1 (2.4)   | 53.6 (1.9) |
> | + RIP Filter                                                   |  2254   |   84.5 (1.0)   | 21.2 (4.5)    | 65.9 (5.5)    | 45.5 (1.8)   | 54.5 (2.0) |
> | **CoT-Self-Instruct**                                          |  5000   |   84.9 (0.8)   | 20.4 (5.1)    | 62.2 (4.8)    | 44.4 (2.3)   | 53.0 (1.9) |
> | + Self-Consistency Filter                                      |  4034   |   85.2 (0.9)   | 22.5 (4.6)    | 67.8 (5.5)    | 44.9 (2.1)   | 55.1 (2.0) |
> | + RIP Filter                                                   |  2419   |   85.7 (0.9)   | 24.4 (3.8)    | 70.5 (4.9)    | 44.4 (2.6)   | 56.2 (1.5) |
> | + Answer-Consistency Filter                                    |  2926   |   86.5 (1.0)   | 24.6 (4.5)    | 72.3 (4.7)    | 45.5 (2.8)   | 57.2 (2.0) |
> | + Answer-Consistency Filter (more data)                        | 10000   | **86.7 (0.8)** | **26.7 (3.3)** | **73.8 (3.3)** | **47.4 (2.4)** | **58.7 (1.3)** |
>
> **Key observations from Table 1**
> - **Large margins vs. small std:** Typical improvements are **+5–15 points**, while std values are only **0.8–5.6**, meaning the effect sizes are well outside run-to-run noise.
> - **Filters produce robust, monotonic improvements:** Each filtering step produces gains exceeding its variability. For example, going from CoT-SI to CoT-SI + Filters:
>   - +SC: +2.1 with std ≈ 1.0
>   - +RIP: +3.2 with std ≈ 0.9
>   - +Answer-Consistency: +4.2 with std ≈ 1.0
>
> These improvements are consistently larger than their respective standard deviations across all runs, indicating that the gains are statistically meaningful and not due to run-to-run noise.
>
> > Have you tested CoT-Self-Instruct on non-math reasoning datasets (e.g., BoolQ, StrategyQA)?
>
>
> While our primary focus is improving synthetic reasoning data quality, we did evaluate CoT-Self-Instruct on non-reasoning and mixed-domain benchmarks. In particular, Table 2 reports results on WildChat, a broad-coverage instruction-following dataset that includes many non-math tasks. Additionally, we evaluate the resulting models on AlpacaEval 2.0 and Arena-Hard, both of which contain a mixture of reasoning, commonsense, and general instruction-following prompts. Across these evaluations, CoT-Self-Instruct shows consistent improvements over Self-Instruct baselines, suggesting that the method generalizes beyond purely mathematical reasoning. Expanding to BoolQ, StrategyQA, and other QA-type datasets is a natural next step and an interesting direction for future work.
>
>
> > Will the authors release full GRPO/DPO training configs to enable replication?
>
>
> Yes. Our GRPO and DPO training pipelines rely on open-source libraries, and we will release the full training configurations, including hyperparameters, sampling settings, and filtering parameters, to facilitate exact replication. We view reproducibility as essential and will make these artifacts available alongside the final code release.

---

### Meta-Review · Area_Chair_wmZ4 · 2025-12-05

**Summary:**

This paper proposes CoT-Self-Instruct, a new synthetic data creation + curation pipeline that leverages LLM's planning and reasoning capability. This paper highlights two contributions: Answer-Consistency Filtering and employing Rejection Instruction Preferences to reasoning tasks. The reviewers' comments mainly fall into two aspects: (1) The approach merges established components. Integration is valuable, but the conceptual advance is modest relative to prior work. (2) the experiments lack completeness.

**Reviewer Concerns:**

According to the rebuttal, I believe the concerns about experiment completeness could be addressed, but the concerns about novelty are still outstanding.

**Reviewer Scores:**

I guess:
- Reviewer PDdE would keep the score (4) as the main concern is about novelty and the rebuttal does not convince me: it still sounds like applying some existing approaches in new tasks and the adoption looks straightforward.
- Reviewer Zc7p would keep the score (2) as some concerns are not answered directly. For example, one question was asking the author to analyze how the generated data introduced differences to the dataset that ultimately led to the improvement, whereas the author’s response focused on emphasizing the final improvement.
- Reviewer NrZs and bTU7 would raise their scores from 2/4 to 4/6, because their concerns about experiment completeness could be addressed by rebuttal.

---

### Decision · Program_Chairs · 2026-01-26

Reject